# Colonic In Vitro Model Assessment of the Prebiotic Potential of Bread Fortified with Polyphenols Rich Olive Fiber

**DOI:** 10.3390/nu13030787

**Published:** 2021-02-27

**Authors:** Lorenzo Nissen, Flavia Casciano, Elena Chiarello, Mattia Di Nunzio, Alessandra Bordoni, Andrea Gianotti

**Affiliations:** 1CIRI-Interdepartmental Centre of Agri-Food Industrial Research, Alma Mater Studiorum-University of Bologna, Piazza G. Goidanich, 60, 47521 Cesena (FC), Italy; mattia.dinunzio@unibo.it (M.D.N.); alessandra.bordoni@unibo.it (A.B.); andrea.gianotti@unibo.it (A.G.); 2DiSTAL-Department of Agricultural and Food Sciences, Alma Mater Studiorum-University of Bologna, Piazza G. Goidanich, 60, 47521 Cesena (FC), Italy; flavia.casciano2@unibo.it (F.C.); elena.chiarello2@unibo.it (E.C.)

**Keywords:** FOS, MICODE, microbiota, olive byproduct, prebiotic, VOCs, qPCR, 16S-rDNA MiSeq

## Abstract

The use of olive pomace could represent an innovative and low-cost strategy to formulate healthier and value-added foods, and bakery products are good candidates for enrichment. In this work, we explored the prebiotic potential of bread enriched with Polyphenol Rich Fiber (PRF), a defatted olive pomace byproduct previously studied in the European Project H2020 EcoProlive. To this aim, after in vitro digestion, the PRF-enriched bread, its standard control, and fructo-oligosaccharides (FOS) underwent distal colonic fermentation using the in vitro colon model MICODE (multi-unit colon gut model). Sampling was done prior, over and after 24 h of fermentation, then metabolomic analysis by Solid Phase Micro Extraction Gas Chromatography Mass Spectrometry (SPME GCMS), 16S-rDNA genomic sequencing of colonic microbiota by MiSeq, and absolute quantification of main bacterial species by qPCR were performed. The results indicated that PRF-enriched bread generated positive effects on the host gut model: (i) surge in eubiosis; (ii) increased abundance of beneficial bacterial groups, such as Bifidobacteriaceae and Lactobacillales; (iii) production of certain bioactive metabolites, such as low organic fatty acids; (iv) reduction in detrimental compounds, such as skatole. Our study not only evidenced the prebiotic role of PRF-enriched bread, thereby paving the road for further use of olive by-products, but also highlighted the potential of the in vitro gut model MICODE in the critical evaluation of functionality of food prototypes as modulators of the gut microbiota.

## 1. Introduction

Currently, the exploitation of byproducts from industrial processing of natural feedstocks is a fundamental requisite for sustainability and to contrast pollution generated by synthetic the production of compounds. Polyphenol-Rich Fiber (PRF) is a defatted olive pomace byproduct obtained from olive oil transformation, which was previously exploited under the aegis of H2020 EcoProlive to produce new functional ingredients for bakery foods, thereby increasing their nutritional value [1]. Considering the use of PRF as an innovative and low-cost strategy to formulate healthier and value-added foods, its impact on human colon microbiota and putatively prebiotic potential deserve attention. In fact, from the latest definition of prebiotics [2], it is clear that molecules other than fructo- or galacto-oligosaccharides could aspire to a prebiotic claim. Prebiotics have essential features [3] expressed towards the amelioration of host health, such as the modulation of the microbiota-fostering beneficials while relenting pathogens, as well as the production of microbial compounds, which, in turn, are good for the host, such as principal SCFAs [4,5] or minor MCFAs [6]. By definition, prebiotics are degraded by colon microbes and contribute to the modulation of the whole microflora; for example, fostering the growth of commensals other than probiotics, such as *Bacteroides* spp. [7], related to microbial eubiosis, and thus to host health [8]. Similarly, a prebiotic is directed nonspecifically to the modulation and limitation of those bacterial groups implied in tryptophan catabolism from proteolytic fermentation, whose compounds’ hallmarks have negative impact on the host, such as BCFAs [9] and some indoles [10]. The action of a prebiotic in regard to colon microbiota is, therefore, wider than expected, and the need for a tool enabling study of the complex microbial ecology beneath is a must. In vitro gut models are considered a proper solution because they can explain the impact of prebiotics on human gut microbiota, focusing on the shift in the core microbial groups and that of selected species and their metabolites, assaying diversity, richness, composition, and abundance in the community over time [11]. In this work, for the first time, we are presenting Multi-Unit In vitro Colon Model (MICODE) a versatile in vitro colon model, that we are introducing in the version simulating the proximal colon, which is able to resemble the microbe-driven colon fermentations, as happens in vivo. We used MICODE with fecal donations from three healthy donors to study the prebiotic potential of PRF-enriched breads by an interomic approach, which couples microbial genomics and metabolomics. Standard breads were enriched with 4% (*w/w*) PRF and underwent in vitro gastro-duodenal digestion, and digested fractions were used to unveil and substantiate the capacity of PRF-enriched bread to have prebiotic potential through the study of ecological indicators, such as: (i) microbial biodiversity, (ii) microbial eubiosis, (iii) prebiotic activity, (iv) the production of desirable compounds, such as SCFAs and MCFAs, (v) modulation of detrimental compounds, such as BCFAs and Indoles, (vi) presence of health-related volatiles, and (vii) reduction in bacterial groups implied in proteolytic fermentations.

## 2. Materials and Methods

### 2.1. Fecal Donors

Fecal donors, two females and one male, were used, in good health and aged between 30 and 45 y. Donors did not undergo antibiotic treatment for at least 3 months prior to stool collection, did not intentionally consume pre- or probiotic supplements prior to experiment, and had no history of bowel disorders. Additionally, the donors were normal weight, not smokers, not chronically assuming any drug, and not alcoholic drink consumers. Two were omnivores and one was vegetarian. The three healthy donors were told of the study’s aims and procedures and gave their verbal consent for their fecal matter to be used for the experiments, in agreement with the ethics procedures required at the University of Bologna.

### 2.2. Materials

Chemicals, solvents, and enzymes for in vitro digestion and batch culture fermentation were of the highest analytical grade and were purchased from Sigma-Aldrich (St. Louis, MO, USA) and Carlo Erba Reagents (CEDEX, Val de Reuil, FR), unless otherwise stated. Reagents for molecular biology (PCR and qPCR), as well as kits for DNA extraction and genetic standard purifications, were purchased from Thermo Fisher Scientific (Waltham, MA, USA).

### 2.3. Experimental Bread and Controls

Experimental bread prototypes were previously prepared and characterized [12,13]. Briefly, 4% (*w/w*) of PRF was added to baker’s yeast-leavened breads, and the PRF-added breads (Eco 4%) were compared to their corresponding controls, i.e., the same bread without PRF (Eco 0%).

### 2.4. In vitro Gastric and Duodenal Digestion

The experimental breads underwent in vitro digestion. Briefly, the digestion process was performed on 5 g of bread for 245 min (2 min of oral, 120 min of gastric and 120 min of intestinal digestion) at 37 °C, according to the INFOGEST standardized protocol [14]. During in vitro digestion, several consecutive enzymatic treatments took place by the addition of simulated saliva (containing 75 U/mL α-amylase), simulated gastric juice (containing 2000 U/mL pepsin) at acid pH, and simulated pancreatic juice (containing 10 mM bile and 100 U/mL pancreatin) at neutral pH. After digestion, the resulting solutions were frozen at −80 °C until further in vitro colonic fermentation.

### 2.5. Fecal Batch-Culture Fermentation and Samples Collection

Colonic fermentations were conducted on fructo-oligosaccharides (FOS) from chicory (Sigma-Aldrich, St. Louis, MO, USA), used as the prebiotic positive control, and digested experimental breads. Fermentations were done using an in vitro gut model, Multi-Unit Colon Model (MICODE), obtained through the assembly of Minibio Reactors (Applikon Biotechnology BV, Delft, NL) and controlled by Lucullus PIMS software (Applikon Biotechnology BV, NL). Bioreactors were autoclaved at 121 °C and −1 bar for 15 min and, once cooled, aseptically filled with 90 mL of anaerobic pre-sterilized basal nutrient medium according to previous publications [15]. Basal medium (BM) contained (per L): 2 g peptone, 2 g yeast extract, 0.1 g NaCl, 0.04 g K_2_HPO_4_, 0.04 g KH_2_PO_4_, 0.01 g MgSO_4_·7H_2_O, 0.01 g CaCl_2_·6H_2_O, 2 g NaHCO_3_, 2 mL Tween 80, 0.05 g Hemin dissolved in 1 mL of 4 M-NaOH, 10 mL vitamin K, 0.5 g L-cysteine HCl, and 0.5 g bile salts (sodium glycocholate and sodium taurocholate). The medium was adjusted to pH 7.0 before autoclaving and 2 mL of 0.025% (*w/v*) resazurin solution were added afterwards, once the media was cooled. Fermentation vessels were filled aseptically with 90 mL of BM and the bioreactor headplates were mounted on previously sterilized and calibrated sensors, i.e., pH and Dissolved Oxygen (DO_2_) sensors. Anaerobic condition (0.0% *w/v* of DO_2_) in each bioreactor was obtained after about 30 min flushing with filtered O_2_-free N_2_ through the mounted-in sparger of Minibio reactors (Applikon Biotechnology BV, NL), and was constantly maintained over the experiment. Temperature was set at 37 °C and stirring at 300 rpm, while pH was adjusted to 5.75 and kept throughout the experiment with the automatic addition of filtered NaOH or HCI (0.5 M), to mimic the conditions located in the proximal region of the human large intestine. Once the exact environmental settings were reached, each of the four vessels were aseptically injected with 10 mL of fecal slurry (10% *w/v* of human feces) to a final concentration of 1% (*w/v*), and then with 1 g of the appropriate substrate/treatment (FOS, digested Eco4% and Eco0% breads) for a final concentration of 1% (*w/v*) [16]. The fourth vessel was set to be free of substrate. Fresh human fecal samples were collected in an anaerobic jar, maintained at 4 °C and processed within 1 h. Fecal slurry was prepared by homogenizing the feces in pre-reduced phosphate-buffered saline (PBS) [9]. Batch cultures were run under these controlled conditions for a period of 24 h, during which samples were collected at four timepoints (0, 5, 10, and 24 h). Sampling was performed with a dedicated double-syringe-filtered system connected to a float drawing from the bottom of the vessels without perturbing or interacting with the bioreactor’s ecosystem. To guarantee a close control, monitoring and recording of fermentation parameters, the software Lucullus 3.1 (PIMS, Applikon Biotechnology BV, NL) was used. This also allowed the stability of all settings to be maintained during the experiment. Fermentations were conducted in duplicate independent experiments, using a new pool of feces for each, from the same three healthy donors. Beside the experimental breads (Eco 0% and Eco 4%) and the positive control (FOS), a blank (basal medium and 1% fecal slurry only) was also included. Parameter trends of the experiments are reported in the Appendix A.

### 2.6. Pipeline of Analytical Activities

Samples of the different timepoints were used for qPCR and SPME GC/MS or at 24 h for the 16S-rDNA MiSeq analyses. Specifically, microbial DNA extraction was conducted just after sampling. DNA samples and GC/MS samples were then stored at −80 °C. Technical replicas of analyses were conducted in duplicate for SPME GC/MS and 16S-rDNA MiSeq, and in triplicate for qPCR. Statistical analyses were also reported below, in detail.

#### 2.6.1. DNA Extraction, Amplification and Sequencing

DNA was extracted from each sample at the baseline and the endpoint using the Purelink Microbiome DNA Purification Kit (Invitrogen, Thermo Fisher Scientific, Carlsbad, CA, USA). Nucleic acid purity was tested on BioDrop Spectrophotometer (Biochrom Ltd., Cambridge, UK). The bacterial diversity was obtained by the library preparation and sequencing of the 16S rDNA gene. The following two amplification steps were performed: an initial PCR amplification using 16S locus-specific PCR primers (16S-341F 5′-CCTACGGGNGGCWGCAG-3′ and 16S-805R 5′-GACTACHVGGGTATCTAATCC-3′) and a subsequent amplification integrating relevant flow-cell-binding domains (5′-TCGTCG GCAGCGTCAGATGTGTATAAGAGACAG-3′ for the forward primer and 5′-GTCTCGTGGGCTCGGAGATGTGTATAAGAGACAG-3′ for the reverse overhang), and unique indices selected among those available Nextera XT Index Kits were combined according to manufacturer’s instructions (Illumina Inc, San Diego, CA, USA). Both input and final libraries were quantified by Qubit 2.0 Fluorometer (Invitrogen, USA). In addition, libraries were quality-tested by Agilent 2100 Bioanalyzer High Sensitivity DNA assay (Agilent technologies, Santa Clara, CA, USA). Libraries were sequenced in a MiSeq (Illumina Inc, USA) in the paired end with 300-bp read length [17]. Sequencing was conducted by IGA Technology Service S.r.l. (Udine, Italy).

#### 2.6.2. Sequence Data Analysis

Reads were de-multiplexed based on Illumina indexing system. Sequences were analyzed using QIIME 1.5.0 [18]. After filtering based on read quality and length (minimum quality = 25 and minimum length = 200), Operational Taxonomic Units (OTUs) defined by a 97% of similarity were picked using the Uclust v1.2.22 q method [19] and the representative sequences were submitted to the RDP classifier [20] to obtain the taxonomy assignment and the relative abundance of each OTU using the Greengenes 16S rRNA gene database [21]. Alpha- and beta-diversity analyses were performed using QIIME 1.5.0 [18].

#### 2.6.3. Enumeration of Bacterial Groups

Changes in Eubacteria kingdom, *Lactobacillales* order, *Bifidobacteriaceae*, *Enterobacteriaceae*, and *Clostridiaceae* families, and *Escherichia coli* species were also assessed by quantitative polymerase chain reaction (qPCR) and SYBR Green I chemistry [22] targeting small fragments of monocopy, or multicopy genes by degenerated or specific primer pairs, previously amplified by high-fidelity DNA polymerase (Invitrogen Platinum SuperFi II DNA Polymerase, Thermo Fischer Scientific, USA). Extraction of bacterial DNA was obtained with Pure Link Microbiome DNA Purification Kit (Invitrogen, USA). Genetic standards were prepared from relative PCR amplicons of the target bacterial species, using GeneJet Genomic DNA purification kit (Thermo Fisher Scientific, USA), as described previously [22,23]. For each of the targets, general qPCR reactions were set as follows: a holding stage at 98 °C for 6 min, and a cycling stage made of 95 °C for 20 sec and 60 °C for 60 sec, repeated for 45 times, followed by melting curve analysis. Quantifications were made by a RotorGene 6000 (Qiagen, Hilden, Germany) with a five-point standard of the given amplicon, separately. Reactions were prepared with 1 ng of DNA, 2× Power-Up SYBR Green (Thermo Fisher Scientific, USA) and 250 nM of each primers (Eurofins Genomics, Ebersberg, Germany). Details of primer pairs for PCR and qPCR are supplied as Appendix A. All results were expressed as mean values obtained from triplicates from two independent experiments.

#### 2.6.4. Volatilome Analysis

Volatile organic compound (VOCs) evaluation was carried out on an Agilent 7890A Gas Chromatograph (Agilent Technologies, Santa Clara, CA, USA) coupled to an Agilent Technologies 5975 mass spectrometer operating in the electron impact mode (ionization voltage of 70 eV) equipped with a Chrompack CP-Wax 52 CB capillary column (50 m length, 0.32 mm ID) (Chrompack, Middelburg, The Netherlands). The Solid Phase Micro-Extraction (SPME) GC-MS protocol and the identification of volatile compounds were done according to previous reports, with minor modifications [23,24]. Briefly, 3 mL of vessel content or fecal slurry were placed into 10-mL glass vials and added to 10 μL of 4-methyl-2-pentanol (final concentration, 4 mg/L), as the internal standard. Samples were then equilibrated for 10 min at 45 °C. SPME fiber, coated with carboxen-polydimethylsiloxane (85 μm), was exposed to each sample for 40 min. Preconditioning, absorption, and desorption phases of SPME–GC analysis, and all data-processing procedures were carried out according to previous publications [23,24]. Briefly, before each head space sampling, the fiber was exposed to the GC inlet for 10 min for thermal desorption at 250 °C in a blank sample. The samples were then equilibrated for 10 min at 40 °C. The SPME fiber was exposed to each sample for 40 min, and finally the fiber was inserted into the injection port of the GC for a 10 min sample desorption. The temperature program was: 50 °C for 1 min, then programmed at 1.5 °C/min to 65 °C, and finally at 3.5 °C/min to 220 °C, which was maintained for 25 min. Injector, interface, and ion source temperatures were 250, 250, and 230 °C, respectively. Injections were carried out in split-less mode and helium (3 mL/min) was used as a carrier gas. Identification of molecules was carried out by searching mass spectra in the available databases (NIST 11 MSMS library and the NIST MS Search program 2.0 (NIST, Gaithersburg, MD, USA). Each VOC was relatively quantified in percentage. Besides, in samples prior to in vitro colonic fermentation (baseline) (Appendix A) the main microbial metabolites related to prebiotic activity were also absolutely quantified in mg/Kg. For these latter compounds, samples at the endpoint (24 h) were compared to the baseline and values were expressed as shifts. All results were expressed as normalized mean values obtained from duplicates in two independent experiments.

#### 2.6.5. Statistical Analysis

For the sequencing data analysis, the QIIME pipeline version 1.5.0 [18] was used. Within-community diversity (alpha diversity) was calculated using observed OTUs, Chao1 Shannon, Simpson, and Good’s coverage indexes with 10 sampling repetitions at each sampling depth. Student’s *t*-test was applied to compare the latest sequence/sample values of different treatments within an index. Analysis of similarity (ANOSIM) and the ADONIS test were used to determine statistical differences between samples (beta diversity) following the QIIME compare_categories.py script and using weighted and unweighted phylogenetic UniFrac distance matrices. Principal Coordinate Analysis (PCoA) plots were generated using the QIIME beta diversity plots workflow [17]. For the rest of the data analyses, Statistica version 8.0 (Tibco Inc., Palo Alto, CA, USA) was used. For the microbiota, the qPCR analysis, and the volatilome one-way ANOVA was used to determine differences between fermentation treatments (blank control, FOS, Eco0%, and Eco4%) at similar timepoints (0, 5, 10 or 24 h), followed by the Tukey’s Honestly Significant Difference (HSD) *post hoc* test. Principal Component Analysis (PCA) was used to evidence discrimination between communities among treatments and applied to a normalized dataset of significant bacterial species (Bonferroni test *p* < 0.05), while PCA and multi-variate ANOVA (MANOVA) were applied to super-normalized datasets of chemical classes of the volatilome. Lastly, a Spearman rank correlation dataset was generated, coupling two independently normalized datasets of the relative quantifications of main metabolites related to prebiotic activity and the microbiota at the species level, then expressed as a two-way joining heatmap with Pearson dendrograms. When the results were presented as shift, we considered them with respect to a baseline of values that was obtained from the analyses of the fecal slurry diluted in PBS and in the BM with the supplementation of the sample to be tested (FOS, Eco0%, and Eco4%).

## 3. Results and Discussion

### 3.1. Quality Controls for the Validation of MICODE

To validate the MICODE in vitro model in the version of fecal batch of the human proximal colon, we chose to monitor and check some parameters as quality controls, other than the trends of the experimental conditions that were plotted over the experiments by Lucullus 3.1 (Applikon Biotechnology BV, The Netherlands) (Appendix A). Quality controls were both related to metabolites and microbes at the end of fermentations, and in comparison to the baseline. Specifically: (i) The *Firmicutes*/*Bacteroidetes* ratio (F/B), which is related to health and disease [25], was maintained at a low level, confirming the capacity to simulate a healthy in vivo condition for 24 h. (ii) The presence of *Archea* (e.g., *Methanobrevibacter smithii*) was retained from stools throughout the experiment in each vessel and repetition, indicating that the environmental conditions were prolonged all over. In fact, *Archea* are renowned for their sensitivity to environmental stressors [26]. (iii) The alpha diversity indices of the microbiota were kept similar. For example, the Good’s rarity index was unchanged (*p* > 0.05), indicating the ability of MICODE to support the growth of rare and fastidious species, while the Observed OTUs richness index scored more than 400 OTUs at the endpoints. (iv) The paradigm of prebiotics was confirmed; in fact, a massive probiotic and SCFA increase and a minimal depletion of enteropathogens were recorded when FOS was applied on MICODE. (v) Each GC/MS analysis had quantified some stool-related compounds (Urea, 1-Propanol, and Butylated hydroxy toluene), that ranged across the complete chromatogram and were adsorbed at the same retention times. Quality controls on the biodiversity of the microbiota undergoing in vitro colonic fermentation were firstly introduced by Takagi and colleagues to confirm that the mixture of microorganisms growing in the in vitro system truly represented the taxonomic diversity of the human microbiota [27].

### 3.2. Changes in Fecal Bacterial Alpha and Beta Diversities

The diversity of gut microbiota within a community was measured with alpha diversity indices (Appendix A), such as the number of observed OTUs for richness, the Chao1 estimator for abundance, the Shannon entropy for evenness, the Simpson index for dominance, and the Good’s measure for rarity. Along alpha diversity, beta diversity with Bray–Curtis analysis and ANOSIM was evaluated to consider differences among samples (Appendix A). At the endpoint, there were differences among the substrates with respect to the baseline for any alpha diversity index except Simpson’s and Good’s. Although the Eco0% and Eco4% did not differ from each other, all indices decreased with respect to the baseline (*p* < 0.05). FOS was the substrate that reduced all the indices the most. For example, the decrease in richness and evenness was explained by the trend of dominance that indicated that some OTUs were overwhelming the others, reducing both richness and the uniform distribution of bacteria. Otherwise, the Simpson index was reduced but not significantly (*p* > 0.05). This effect was due to the ability of FOS to foster just some species, e.g., probiotics, and make them dominant over the microbiota, thus reducing overall alpha diversity. Eco4% had a mitigated effect on diversity reduction; for example, while FOS reduced the observed OUT index from the baseline by 2.2-fold (*p* < 0.05), Eco4% reduced it by just 1.6-fold (*p* < 0.05). Besides, considering the evenness of the microbiota, the Shannon index of FOS and Eco4% was 1.8- and 1.4-fold smaller that the baseline (*p* < 0.05). The lower capacity of ECO4% to reduce these indices compared to FOS suggested that it had a wider range of bioactivity on more bacteria targets. The unchanged Good’s diversity index from the baseline throughout the fermentations indicated that MICODE maintained a stable ecological condition even for rare bacterial species. When the bacterial diversity between samples (beta diversity) was examined, ANOSIM clustering was shown on a time basis, clustering the baseline cases near the donors and separately from the endpoint cases, as demonstrated by principal coordinate analysis (PCoA) based on an unweighted (qualitative) phylogenetic UniFrac distance matrix. Moreover, beta diversity PCoA included the nearby replicas of cases, indicating scarce experimental variations and significant ANOSIM (*p* < 0.01). A reduction in alpha diversity indices was reported in a similar work conducted with a similar in vitro model [16], and could point out the limit of single-batch in vitro models.

### 3.3. Fecal Bacterial Relative Abundance at the Phylum Level

The total sequence reads used in this study were classified into ten phyla, two other than the previous, and one was unassigned (Table 1). In any tested case, the core microbiota was represented by five main phyla, i.e., *Firmicutes*, *Bacteroidetes*, *Actinobacteria*, *Proteobacteria*, and *Verrucomicrobia*, in a descending proportional order. These phyla, along with the less represented *Euryarcheota*, were subject to significant changes over colonic fermentation in comparison to the baseline (*p* < 0.05). To evidence the prebiotic potential of Eco4%, it is important to stress the trend of the ratio *Firmicutes/Bacteroidetes* (F/B) over time. In fact, this ratio indicates an eubiosis of the microbiota when ranging around 1.5, and a dysbiosis when more than 2, leading to intestinal syndromes [25,28]. In this study, at baseline, fecal samples recorded an F/B of around 1.6, indicating the healthy condition of the donors, and this ratio was kept similar after fermentation either with FOS (1.7) or Eco4% (1.3), while it was 2.3-fold higher when colonic fermentation was conducted with Eco0% (*p* < 0.05). These findings confirmed the positive role of olive oil polyphenol compounds observed in human studies in increasing *Bacteroidetes* and/or reducing the F/B ratio [29].

### 3.4. Discriminant Microbiota Populations at the Species Level

At the lowest taxonomic level, 189 distinct bacterial OTUs were constructed and assigned (cutoffs 0.001%). Of these, 97 were identified at the baseline, while 54, 78, and 72 were identified at the endpoint of FOS, Eco0% and Eco4% fermentations, respectively. A dataset of 62 significant bacterial OTUs (Appendix A) was generated after ANOVA (*p* > 0.05) and used to perform untargeted multivariate analysis by PCA. Successful discrimination of the variables among the substrates was achieved at the endpoint (Figure 1). In detail, major descriptors for FOS were *Bifidobacterium adolescentis*, *Bif. Bifidum*, *Akkermanisa muciniphila*, and *Roseburia faecis*. Those for Eco0% were the *Bacteroides cellulosyliticus*, *Dorea formicigerans*, and *Bilophila wadsworthia*. Those for Eco4% were *Megasphera elsdenii*, *Parabacteroides distasonis*, *Entorococcus durans*, *Bif. longum*, *Faecalibacterium prausnitzii*, *Lactobacillus plantarum*, *B. massiliensis*, *B. caccae*, and *B. uniformis*. The OTUs with the highest increases (*p* < 0.05) at the endpoint were *Bif. adolescentis*, *Bif. longum*, and *Lach. pectinoschiza* after FOS fermentation, and *M. elsdenii*, *Ent. durans*, *P. distansonis*, *B. massiliensis*, *L. plantarum*, and *Bif. longum* after Eco4% fermentation. In contrast, variables with maximum reductions (*p* < 0.05), either after FOS or Eco4%, were *F. prausnitzii*, *B. vulgatus*, *Ruminococcus gnavus*, *Citrobacter freundii*, *E. albertii*, and *Bil. wadsworthia*. Thus, from our results, even at the depth of the species level, it was possible to highlight the prebiotic potential of Eco4% that, similarly to FOS, fostered probiotic bacteria as well as beneficial bacteria, such as the SCFAs-producer *M. elsdenii* [30], MCFAs- and sphingolipids-producer *B. massiliensis* [31], succinate-producer *P. distasonis* [32], and competitive excluders *Lactobacillales* [33], as *L. plantarum* [34] and *E. durans* [35]. Moreover, Eco4% had the highest loads in beneficial SCFAs-producer *F. prausnitzii* [36] and fiber-degrading *B. caccae* [37]. Fitting with the concept of prebiotics, Eco4% was even able to limit and contrast the growth of opportunistic (*Citrobacter freundii*) [38] and close relative pathogenic species (*Escherichia albertii*) [39], as well as that of species related to metabolic syndrome enterotypes, such as *R. gnavus* [40] and sulphate-producer *Bil. wadsworthia* [41].

### 3.5. Changes in Selected Fecal Bacterial Populations Measured with qPCR

Changes in *Eubacteria* kingdom, *Lactobacillales* order, *Bifidobacteriaceae*, *Enterobacteriaceae*, and *Clostridiaceae* families, and E. coli species were also assessed by qPCR (Table 2). At the early timepoint (5 h), no significant changes were found among all cases and bacterial targets (*p* > 0.05). At the intermediate timepoint (10 h), *Bifidobacteriaceae* and *Lactobacillales* significantly increased in numbers for FOS and Eco4% fermentations, while *Clostridiaceae* increased for Eco0% (*p* < 0.05). At the endpoint (24 h), almost all bacterial targets significantly changed in abundance (*p* < 0.05). For example, FOS, along with Eco4%, had increased numbers of total *Eubacteria*, *Bifidobacteriaceae*, and *Lactobacillales*. Eco4%, along with Eco0%, even recorded an increase in *Clostridiaceae*. Exclusively, Eco0% caused *Enterobacteriaceae* and *E. coli* surging. Besides this, Eco4% contributed to the significant reduction in *E. coli*. Our results are comparable to those obtained in the literature by similar investigations in similar colon models [15,16], and those of FOS and Eco4% are strictly limited to the concept of prebiotics, for which a compound must foster the growth of beneficial and probiotic bacteria (*Bifidobacteriaceae* and *Lactobacillales*) while simultaneously reducing that of opportunistic and pathogenic bacteria (*Enterobacteriaceae* and *E. coli*). The sole difference between FOS and Eco4% was the divergent shift recorded at the endpoint for *Clostridiaceae*. Similar features were reported previously in research studying the impact of olive oil on gut microbiota [42].

### 3.6. Volatilome Analysis through SPME GC/MS

Through SPME GC-MS, among 24 duplicated cases (*n* = 48), 161 molecules were identified with more than 80% of similarity with NIST 11 MSMS library and the NIST MS Search program 2.0 (NIST, Gaithersburg, MD, USA). On average, 92 were relatively quantified at the baseline, while 120 were quantified during the 24 h of experiments at different timepoints (Appendix A). For a landscape description of the volatilome, a dataset of 49 significant molecules (ANOVA at *p* < 0.05) was generated, then sorted and super-normalized by chemical classes of VOCs, i.e., organic acids, aldehydes, ketones, and alcohols and indoles. Organic acids and indoles are discussed in paragraph 3.7, as main microbial metabolites related to prebiotic activity, while, from each dataset of the other classes, multivariate analyses, such as untargeted Principal Component Analysis (PCA) and targeted MANOVA (*p* < 0.01) was achieved to address the specific contributes to VOCs production by the independent variables. Super-normalization of the dataset was essential to unveil the effect of those compounds that are less volatile than others and could be underrepresented, as well as to avoid comparing one chemical class to another [43]. Aldehydes are a result of microbial fermentation and lipid oxidation, as well as the transformation of ethanol [44]. Certain aldehydes are health-promoters, because they contribute positively to cell homeostasis and microbiota eubiosis, such as Indole-3-aldehyde [45], while most are detrimental, being cytotoxic at a low threshold, such as Acetaldehyde [46]. A PCA of 10 statistically significant aldehydes showed distributed cases on the plot, separating fermentation with FOS and Eco4% from each other and from the baseline (Figure 2A). the main descriptor of fermentation with FOS was Butanal, 2-methyl (*p* < 0.01), while those of Eco4% were Benzeneacetaldehyde, Nonanal, and 2-Nonenal, (E), with the latter principally produced at the endpoint (24 h) (*p* < 0.01) (Appendix A). 2-Nonenal, (E) was reported to limit the growth of several intestinal pathogens at a very low concentration [47]. It is conceivable that this resulted from the degradation of Palmitoleic acid [48], which is one of the main fatty acids in olive oil. During colon fermentation, many ketones are produced; considering their bioactive attributes, some are desirable, such as the ketones bodies [49], others, such as acetone, are unwanted, because they could be toxic for the host [50]. The PCA of 13 statistically significant ketones distributed cases on the plot, separating the substrates from each other and from the baseline (Figure 2B). Descriptors of fermentation with Eco4% were Hexanone-5-methyl, 2-Butanone and Acetophenone, largely produced at the endpoint (*p* < 0.01). The main descriptor of fermentation with FOS was 2-Butanone, 4-hydroxy (*p* < 0.01), while the principal of Eco0% was acetone (*p* < 0.05) (Appendix A). Acetophenone deserves attention, since it acts as antimicrobial to different Gram-negative bacteria [51], and its *N*-substitute derivates have been proposed as a therapeutic approach in diabetes [52]. In our experimental condition, it probably derived from the bacterial deconjugation of polyphenols, where Eco4% is rich. A bacterial group implied in such action is *Lactobacillales* [53], which was increased after Eco4%. Alcohols are essential compounds of the fermentation of dietary polysaccharides conducted by the colon microbiota [54]. PCA of 13 statistically significant alcohols distributed cases on the plot, separating fermentation with FOS and Eco4% from each other and from the baseline (Figure 2C). From our results, the contribution to alcohol production from the control samples remains undiscriminated (*p* > 0.01), while the descriptor of fermentation with FOS was mainly Ethyl alcohol (*p* < 0.01), and those for Eco4% were phenethyl alcohol, 1-hexanol and 1-Pentanol, mainly produced at the late timepoints (*p* < 0.01) (Appendix A). It is reported that 1-Pentanol is associated with the consumption of old grains, which have anti-inflammatory and prebiotic activity [24].

### 3.7. Changes in Main Microbial Metabolites Related to Prebiotic Potential

To analyze the main changes in volatile microbial metabolites related to prebiotic potential, we considered the shift in loads from the baseline to the endpoint (24 h) of the fermentation of 13 selected VOCs with renowned bioactivity in humans (SCFAs, MCFAs, BCFAs, Indole and Skatole) as follows: (a) each single compound was normalized (mean centering method) within its dataset, which included cases from different type of sample; (b) the baseline dataset (Appendix A) was then subtracted to the endpoint dataset; (c) post-hoc analysis was done to compare the sample productions of a single molecule (Tukey’s HSD, *p <* 0.05). Short Chain Fatty Acids (SCFAs) are essential compounds for the host, the mucosa, and the colon microbiota. From our results (Figure 3A), SCFA concentration increased with FOS and Eco4%, while Acetic and Propanoic acid concentration decreased with Eco0%. The capacity to produce SCFAs was in the order FOS>ECO4%>ECO 0% (*p* < 0.05). A reduction in SCFAs content is linked to a reduced eubiosis of the gut microbiota and a reduced intestinal cell homeostasis [55]. The prebiotic activity of Eco4% could, therefore, be linked to its capacity to foster bacteria, deconstructing fibers and producing SCFAs. In fact, as discussed above, we recorded an increase in *Lactobacillus* spp., *Bifidobacterium* ssp., and *Enterococcus* spp., which are beneficial and are hard-working in terms of SCFAs production [2]. Possibly, this effect could be due to the PRF compounds of Eco4%, as other authors reported that these species are able to hydrolyze oleuropein, a polyphenol rich in olive oil, to form hydroxycortisol acetate [29,56]. Medium-Chain Fatty Acids (MCFAs) have protective effects on glucose homeostasis during high-fat overfeeding and against insulin resistance [57] and are important metabolic biomarkers of dysbiosis associate to Intestinal Bowel Disease (IBD) [28,58]. MCFAs’ concentration increased after fermentation with FOS or Eco4% compared to baseline (Figure 3B and Appendix A). Slight changes were seen after fermentation with Eco0%, indicating that the Eco4% effect was probably related to changes in the colon microbiota more than the fatty acid composition of the bread. FOS fermentation produced the highest loads of any MCFAs tested, except octanoic acid, whose production was 5.7-fold higher in ECO4% than in FOS (*p* < 0.05). Besides, Eco4% produced 5.2- and 8.9-fold more hexanoic and heptanoic acids, respectively (*p* < 0.05), than its control ECO0%. The increased abundance in MCFAs observed in this study could be due to the ability of Eco4% to foster *Bifidobacteriaceae* and commensals *Enterobacteriaceae* and *Bacteroides* spp., as we have previously described. Actually, MCFA production by these three bacterial groups happened during fiber fermentation [59,60]. Branched-Chain Fatty Acids (BCFAs), such as Propanoic acid, 2-methyl, Butanoic acid, 3-methyl, and Pentanoic acid, 3-methyl, are derived from microbial colon protein fermentation and produce NH_3_, phenol and sulphate amines that could result stressful for the host [61]. BCFAs are often used as a biomarker of protein catabolism, with the promoted target used to reduce their concentration and improve health outcomes [62]. Still, little is known about the impact of BCFAs on host health [54]. What is undisputed, however, are the negative consequences of the pro-inflammatory and cytotoxic compounds yielded from the sulfur-containing, basic and aromatic amino acids [54]. From our results, BCFAs (Figure 3C) increased with just Eco0%. Modest increments were seen for Propanoic acid, 2-methyl (Prop2M) and for Pentanoic acid, 3-methyl (Penta3M), when FOS and Eco4% were supplied, respectively. FOS and Eco4%, therefore, were able to reduce Butanoic acid, 3-methyl (Buty3M) similarly. A reduction or an increment, driven by Eco4%, could be seen, in comparison to the surge seen for the three BCFAs made by Eco0%, showing that our food product is shaping the microbiota, fostering the growth of the core microbiota that specializes in the fermentation of fibers, more than that specializing in protein fermentation. The different proportions of fermentable protein and carbohydrates available for the microbiota in the two experimental breads might have suppressed AA fermentation, as already reported by [63]. Since a similar situation was observed after FOS fermentation, another notch was added to the prebiotic potential of Eco4%. Indole and Skatole are two compounds of tryptophan catabolism, deriving from degradation of the proteinaceous portion of the food. Besides tryptophan metabolism by the host, resident microbiota can directly convert tryptophan into indoles, and several different derivatives are formed [64,65]. Whereas Indole is also suggested to have beneficial effects, such as the attenuation of inflammation indicators [66], its bacterial production (*Clostridium* spp. and *Escherichia* spp.) and accumulation is toxic for the host, because it alters the permeability and homeostasis of the mucosa [10]. Once metabolized into Indoxyl sulphate in the liver, it can lead to chronic kidney disease and vascular disease [9,65]. Besides this, the bacterial decarboxylation (*Bacteroides* spp. and *Clostridium* spp.) of tryptophan produces skatole (Indole,3-methyl) which causes the production of inflammatory cytokines [10]. From our results (Figure 3D), the shifts recorded by FOS and Eco4% fermentations compared to baseline indicated a reduction in both Indole and Skatole concentration, while the shift in Eco0% was the opposite. Similar to what happened for BCFAs, the prebiotic potential of Eco4% could be ascribed to PRF addition, which improved the proportions of fermentable protein and carbohydrates in the experimental bread, shaping the microbiota to the advantage of bacterial groups specialized in fibers more than protein fermentation.

### 3.8. Interomics Correlations among Metabolites Related to Prebiotic Potential and the Microbiota

Spearman Rank Correlations (*p* < 0.05), two-joining-way Heatmaps, and Pearson cluster analysis were performed by the comparison of two different normalized datasets, each derived from values of relative quantification (OTUs and VOCs) (Figure 4). The significance of correlations is reported in the Appendix A. From the Pearson dendrograms, three clusters were identified: the first two could strengthen the outputs on prebiotic potential presented over Eco4% fermentations, while the third includes less abundant OTUs and is supposedly less metabolically active in Eco4%. The first cluster included *Bif. adolescentis*, *M. elsdenii*, *Lach. pectinoshiza*, and *Colinsella aerofaciens*, and it was positively correlated to SCFAs and MCFAs abundances, and inversely correlated to Pentanoic acid, 3-methyl and Skatole. What is known is that *M. elsdenii* produces butyrate from acetate or lactate produced from *Bif. adolescentis* [55]. Lactate, which is not an SCFA, is also produced as a result of fermentation, but does not accumulate in the colon because it is used by several SCFA-producing bacteria [30], like *Lach. pectinoshiza*, and *Colinsella aerofaciens*. Another explicative correlation was found in cluster 2, where the reduction in indole content and BCFAs after Eco4% fermentation could be due to the recovered high loads in *E. durans*, while the reduction in skatole content was correlated with well-represented OTUs in Eco4% fermentations, such as *F. prausnitzii* and *A. muciniphila*. Moreover, these three species, along with *P. distasonis* (cluster 1), were positively correlated with octanoic acid that we found after fermentations that were richer in Eco4% than FOS. The increased abundances in MCFAs were correlated in cluster 2 to more commensal *Enterobacteriaceae*, *Bif. longum,* and *Bif. bifidum,* as seen after FOS or Eco4% fermentations, and as explained by other authors [59,60]. Eco4% and FOS were even able to diminish the population of opportunistic *Enterobacteriaceae* and *Desulfovibrionaceae,* as well as the production of indoles and BCFAs; in fact, we found a positive correlation among *Cit. freundii*, *E. albertii* and *Bil. wadsworthia* (cluster 3) with the production of those detrimental compounds.

## 4. Conclusions

Based on the positive results obtained by different prebiotics indicators, our study evidenced that Eco4% bread had a prebiotic potential ascribable to PRF addition. Indeed, Eco4% did not affect eubiosis and did not induce dysbiosis, maintaining a balanced F/B ratio and keeping a similar alpha diversity of the composition of the microbiota over the fermentation period. Other positive indicators were the increased production of SCFAs, MCFAs and the reduction in abundance of harmful BCFAs, indole, and skatole. Furthermore, an increased abundance of probiotic or beneficial species was observed after Eco4%, as well as a decrease in opportunistic or pathogenic species. Lastly, microbiota after Eco4% showed an increase in species related to fiber fermentation, along with a reduction in those associated with proteolytic fermentation. Our results were backed by a logical and clear multivariate statistical approach, able to combine microbial genomics data and microbial metabolomics data in an interomic showcase that visibly demonstrates the cause and effects generated by a certain fiber possessing prebiotic potential. The presented study was performed using MICODE, a robust and versatile in vitro model, that was assessed by a quality control check of different issues, such as the presence, throughout the fermentation period, of *Archea* species, the capacity of FOS to foster probiotics, the of similar observed OTUs in the system, as well as the rare species seen by Good’s index, and, lastly, considering the volatilome, there were several stool-derived compounds kept at the same retention time on any samples. Even if in vivo animal models or diet-intervention studies should be used to fully elucidate the prebiotic potential of Eco4%, as well as to address specific host benefits, the recipient results we have presented are target-effective and should have robustness for pre-clinical applications.

## Figures and Tables

**Figure 1 nutrients-13-00787-f001:**
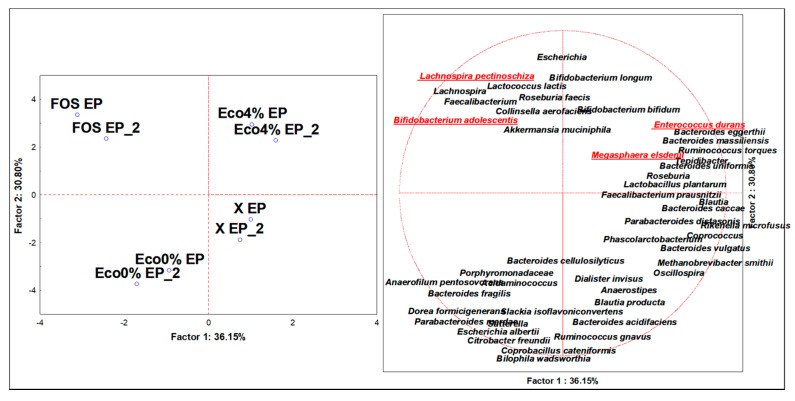
Principal component’s analysis (PCA) of relative abundances (%) of significative (ANOVA *p* < 0.05) assigned Operational Taxonomic Units (OTUs) at the species level, after 24 h (EP = Endpoint; EP_2 = duplicate sample) of in vitro batch culture fermentations inoculated with human feces (*n* = 3 healthy donors) and administrated with FOS, Eco0%, and Eco4% as the substrates and a blank control (X). Variables in red font are the main descriptors of Eco4% or FOS cases.

**Figure 2 nutrients-13-00787-f002:**
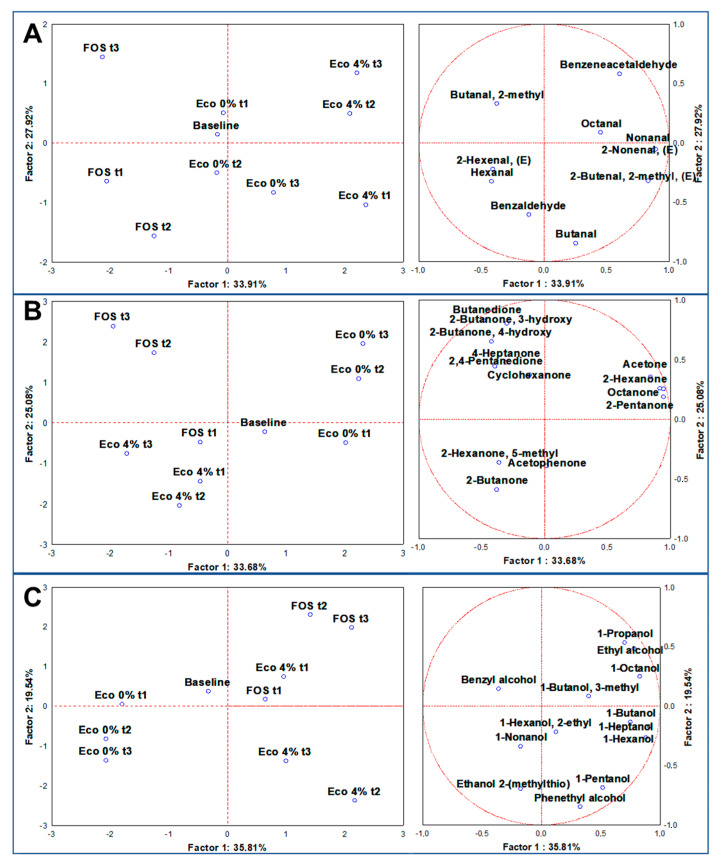
PCAs of the volatilome sorted by chemical classes of significative (ANOVA *p* < 0.05) VOCs, including the baseline and three different timepoints (t1 = 5 h; t2 = 10 h; t3 = 24 h). (**A**) = Aldehydes; (**B**) = Ketones; (**C**) = Alcohols. Left-side diagrams are for PCAs of cases, while right-side diagrams are for PCAs of variables.

**Figure 3 nutrients-13-00787-f003:**
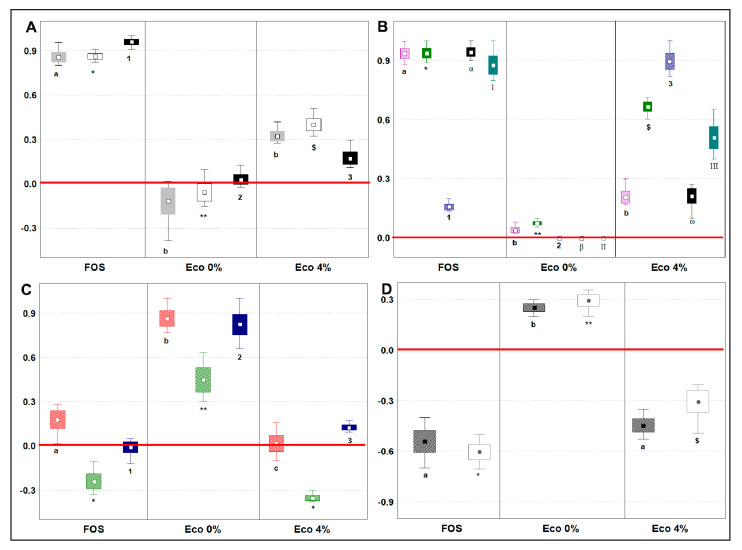
Endpoint changes in main microbial metabolites related to prebiotic activity, expressed as normalized scale from relative abundances with respect to the baseline (red line). The baseline absolute quantifications in mg/Kg are found in the Supplementary Material (Appendix A). Changes were recorded after 24 h of in vitro batch colonic fermentations inoculated with human feces (n = 3 healthy donors) and administrated with FOS, Eco 0%, and Eco 4%. Samples were analyzed in duplicate from two independent experiments (*n* = 4). Box = mean; Rectangles = mean * S.D.; Whiskers = min and max values. Cases with different letters or numbers or symbols among a single independent variable are significantly different according to Tukey’s HSD test (*p* < 0.05). (**A**) Short-Chain Fatty Acids: gray plot = Acetic acid; white plot = Propanoic acid; black plot = Butanoic acid. (**B**) Medium-Chain fatty Acids: Fuchsia plot = Hexanoic acid; green plot = Heptanoic acid; blue stripes plot = Octanoic acid; black plot = Nonanoic; pale blue plot = n-Decanoic acid. (**C**) Branched-Chain Fatty Acids: red plot = Propanoic, 3-metyl acid; green plot = Butanoic, 3-methyl acid; blue plot = Pentanoic, 2-methyl acid. (**D**) Indoles: gray plot = Indole; white plot = Skatole.

**Figure 4 nutrients-13-00787-f004:**
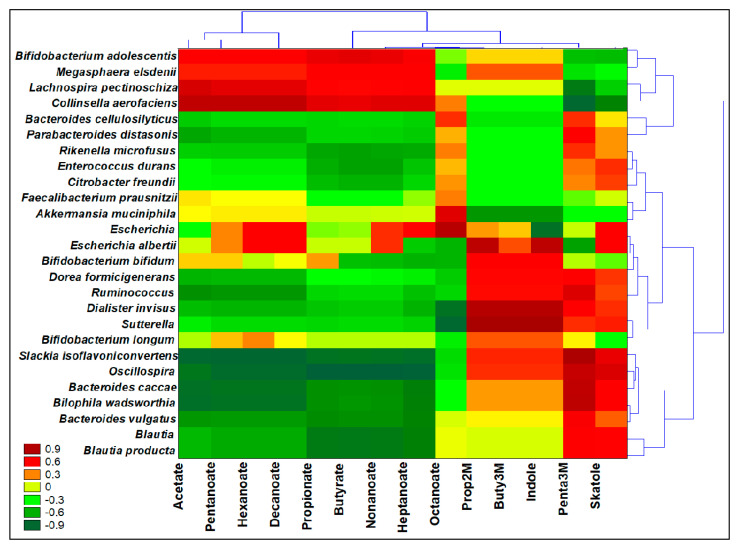
Interomics, Spearman Rank Correlations among main microbial metabolites from the volatilome and ANOVA significant (*p* < 0.05) species OTUs from the microbiota. Prop2M = Propanoic acid, 2-metyl; Buty3M = Butanoic acid, 3-methyl; Penta3M = Pentanoic acid, 3-methyl. Left-side dendrogram identifies by Pearson analysis three major different clusters among bacterial species. Significance of correlations are provided as Appendix A.

**Table 1 nutrients-13-00787-t001:** Changes in bacterial phyla (relative abundances (%)) throughout 24 h in vitro batch culture fermentations inoculated with human feces (*n* = 3 healthy donors) and administrated with fructo-oligosaccharides (FOS), Eco4%, and Eco0% as the substrates.

	Baseline			Endpoint								
*Phylum*				FOS			Eco0%			Eco4%		
*Firmicutes*	54.005	±	0.635 ^b^	41.997	±	1.111 ^a^	63.470	±	0.651 ^c^	41.509	±	0.596 ^a^
*Bacteroidetes*	33.997	±	0.741 ^a^	23.957	±	0.870 ^b^	17.418	±	0.422 ^c^	33.264	±	0.529 ^a^
*Actinobacteria*	7.537	±	0.613 ^a^	27.832	±	1.232 ^b^	6.338	±	0.738 ^a^	16.696	±	1.058 ^c^
*Proteobacteria*	1.762	±	0.193 ^a^	3.577	±	0.544 ^b^	11.628	±	1.344 ^c^	5.571	±	0.798 ^b^
*Verrucomicrobia*	1.775	±	0.218 ^a^	1.175	±	0.128 ^b^	0.207	±	0.065 ^c^	1.910	±	0.478 ^a^
*Euryarchaeota*	0.145	±	0.023 ^a^	0.010	±	0.002 ^c^	0.076	±	0.006 ^b^	0.030	±	0.004 ^c^
*Fusobacteria*	0.009	±	0.001 ^a^	0.001	±	0.000 ^a^	0.001	±	0.000 ^a^	0.084	±	0.014 ^b^
*Synergistetes*	0.011	±	0.002 ^a^	0.001	±	0.000 ^b^	0.007	±	0.002 ^a^	0.001	±	0.000 ^b^
*Tenericutes*	0.009	±	0.001 ^a^	>0.001	±	0.000 ^b^	0.001	±	0.000 ^a^	>0.001	±	0.000 ^b^
*Crenarchaeota*	0.001	±	0.000 ^a^	>0.001	±	0.000 ^b^	>0.001	±	0.000 ^b^	0.001	±	0.000 ^a^
*Bacteria*; Other	0.637	±	0.098 ^a^	0.038	±	0.009 ^c^	0.071	±	0.012 ^b^	0.107	±	0.021 ^b^
*Archaea*; Other	0.004	±	0.001 ^a^	0.001	±	0.000 ^a^	0.002	±	0.000 ^a^	0.001	±	0.000 ^a^
Unclassified	0.030	±	0.005 ^a^	0.005	±	0.001 ^b^	0.017	±	0.005 ^a^	0.018	±	0.006 ^a^
F/B ^1^	1.589	±	0.053 *	1.753	±	0.017 *	3.644	±	0.051 ^§^	1.247	±	0.012 *

^a,b,c^ Letters or, ^*^,^§^ Symbols indicate significant differences within a line by Tukey’s honestly significant differences (HSD) test (*p* < 0.05); ^1^ F/B = Firmicutes/Bacteroidetes. Samples were analyzed at 0 h (baseline) and 24 h. Values are means (%) with S.D.

**Table 2 nutrients-13-00787-t002:** Changes in bacterial populations measured by qPCR on a RotorGene 6000 (Qiagen, Hilden, Germany) with the SYBR Green I chemistry, expressed as mean values in Log_10_ cells/mL.

	Time (h)	*Eubacteria*			*Bifidobacteriaceae*			*Lactobacillales*			*Enterobacteriaceae*			*Escherichia coli*			*Clostridiaceae*		
**FOS**	0	9.16	±	0.16 ^a^	6.77	±	0.11 ^a^	7.36	±	0.09 ^a^	8.60	±	0.07 ^b^	4.08	±	0.03 ^ab^	7.15	±	0.11 ^a^
	5	9.32	±	0.10 ^a^	6.99	±	0.12 ^ab^	7.76	±	0.10 ^ab^	8.62	±	0.13 ^b^	4.40	±	0.10 ^b^	7.22	±	0.04 ^a^
	10	9.77	±	0.09 ^ab^	7.48	±	0.09 ^b^	8.31	±	0.09 ^b^	8.51	±	0.02 ^ab^	4.62	±	0.07 ^b^	7.67	±	0.03 ^ab^
	24	10.09	±	0.28 ^b^	8.81	±	0.23 ^c^	8.79	±	0.11 ^b^	8.05	±	0.06 ^a^	3.62	±	0.07 ^a^	7.34	±	0.30 ^a^
**Eco 0%**	0	9.12	±	0.25 ^a^	6.47	±	0.08 ^a^	7.11	±	0.09 ^a^	8.71	±	0.08 ^b^	4.00	±	0.07 ^ab^	7.11	±	0.11 ^a^
	5	9.00	±	0.11 ^a^	6.71	±	0.09 ^a^	7.65	±	0.11 ^ab^	8.91	±	0.12 ^bc^	4.40	±	0.08 ^b^	7.35	±	0.11 ^a^
	10	9.41	±	0.26 ^a^	6.68	±	0.09 ^a^	7.90	±	0.14 ^ab^	9.14	±	0.11 ^bc^	4.92	±	0.11 ^bc^	7.95	±	0.21 ^b^
	24	9.57	±	0.07 ^ab^	6.27	±	0.08 ^a^	7.71	±	0.11 ^ab^	9.44	±	0.23 ^c^	5.13	±	0.21 ^c^	8.10	±	0.10 ^b^
**Eco 4%**	0	9.02	±	0.12 ^a^	6.77	±	0.10 ^a^	7.24	±	0.10 ^a^	8.40	±	0.06 ^ab^	4.31	±	0.07 ^b^	7.01	±	0.10 ^a^
	5	9.22	±	0.08 ^a^	7.10	±	0.10 ^ab^	7.36	±	0.11 ^a^	8.62	±	0.11 ^b^	4.17	±	0.17 ^ab^	7.23	±	0.20 ^a^
	10	9.70	±	0.09 ^ab^	7.74	±	0.09 ^b^	7.98	±	0.21 ^b^	8.70	±	0.08 ^b^	4.22	±	0.16 ^ab^	7.47	±	0.10 ^ab^
	24	10.03	±	0.20 ^b^	8.55	±	0.15 ^c^	8.80	±	0.14 ^b^	9.16	±	0.19 ^bc^	3.92	±	0.11 ^a^	8.01	±	0.19 ^b^

^a,b,c^ Different letters among a bacterial target indicate significance by Tukey’s HSD test (*p* < 0.05).

## Data Availability

No applicable.

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
