# Peer review of "Colonic In Vitro Model Assessment of the Prebiotic Potential of Bread Fortified with Polyphenols Rich Olive Fiber"

_nutrients, 2021, doi:10.3390/nu13030787_

Round 1
Reviewer 1 Report
The authors should change:
Line 18 Fructo-oligosaccharides should be written form small letter fructo-oligosaccharides
Line 18 and 27 in vitro should be written Italic in vitro
Line 47 micro flora, should be written microflora,
Line 54 and 59 and 64 and 83 and 95 and 98 and 102 and 329 in vitro should be written Italic in vitro
Line 61 in vivo should be written Italic in vivo
Line 519 should be dot . after 63
Author Response
Reviewer 1
----We really thank the Reviewer for the precious work done.
English language and style are fine/minor spell check required
----We have revised the entire submission
The authors should change:
Line 18 Fructo-oligosaccharides should be written form small letter fructo-oligosaccharides
----Revised accordingly
Line 18 and 27 in vitro should be written Italic in vitro
----Revised accordingly
Line 47 micro flora, should be written microflora,
----Revised accordingly
Line 54 and 59 and 64 and 83 and 95 and 98 and 102 and 329 in vitro should be written Italic in vitro
----Revised accordingly
Line 61 in vivo should be written Italic in vivo
----Revised accordingly
Line 519 should be dot . after 63
----Revised accordingly
Reviewer 2 Report
This is an interesting paper in Nutrients. Authors are invited to take into account the following comment and revise minor typo errors found in the manuscript.
Comment:
1) Is the lower Firmicutes/Bacteroidetes ratio (F/B) in systematic way a real quality control parameter of an healthy in vivo condition (Line 261)? If so, the increased abundance of Lactobacillales (beneficial bacterial groups) belonging to Firmicutes as announced by the Authors (Line 23) should have an opposite effect. Moreover, the ref. [25] cited by the Authors reports results and conclusions investigated in one Country population in Europe, which could be different in other Countries.
2) Please revise the following minor typo errors found in the manuscript:
Line 85 Fr ---> FR
Line 138 ... allowed to keep the stability of all settings during...---> allowed the stability of all settings kept during...
Line 249 epressed ---> expressed
Line 307 Experymental ---> Experimental
Line 309 point ---> point out
Lines 346 & 349 Parabcteroides ---> Parabacteroides
Line 360 Oportunistic ---> opportunistic
Line 378 along ---> along with
Line 380 contribute ---> contributed
Line 519 [63] ---> [63].
Author Response
Reviewer 2
----We really thank the Reviewer for the precious work done.
(x) English language and style are fine/minor spell check required
----We have revised the entire submission
This is an interesting paper in Nutrients. Authors are invited to take into account the following comment and revise minor typo errors found in the manuscript.
Comment:
1) Is the lower Firmicutes/Bacteroidetes ratio (F/B) in systematic way a real quality control parameter of an healthy in vivo condition (Line 261)?
----This parameter is a meaning of an equilibrium in the microbiota, which reflects an healthy gut ecology and condition.
Ley RE, Turnbaugh P, Klein S, Gordon JI: Microbial ecology: human gut microbes associated with obesity. Nature. 2006, 444: 1022-1023. 10.1038/4441022a. Citations 7382
If so, the increased abundance of Lactobacillales (beneficial bacterial groups) belonging to Firmicutes as announced by the Authors (Line 23) should have an opposite effect.
----normally and in that case, under the phyla of Firmicutes an increase in Lactobacillales is balanced by a reduction in Clostridiales. Anyhow the portion of this latter in the gut microbiota generally overwhelm the former.
Moreover, the ref. [25] cited by the Authors reports results and conclusions investigated in one Country population in Europe, which could be different in other Countries.
----As donors of the experiment were European, we choose to cite this work, as it is highly cited and one of the more recent. 2017 with 376 citations
2) Please revise the following minor typo errors found in the manuscript:
Line 85 Fr ---> FR
----Revised accordingly
Line 138 ... allowed to keep the stability of all settings during...---> allowed the stability of all settings kept during...
----Revised accordingly
Line 249 epressed ---> expressed
----Revised accordingly
Line 307 Experymental ---> Experimental
----Revised accordingly
Line 309 point ---> point out
----Revised accordingly
Lines 346 & 349 Parabcteroides ---> Parabacteroides
----Revised accordingly
Line 360 Oportunistic ---> opportunistic
----Revised accordingly
Line 378 along ---> along with
----Revised accordingly
Line 380 contribute ---> contributed
----Revised accordingly
Line 519 [63] ---> [63].
----Revised accordingly